# Phytotoxic Responses of Soybean (*Glycine max* L.) to Botryodiplodin, a Toxin Produced by the Charcoal Rot Disease Fungus, *Macrophomina phaseolina*

**DOI:** 10.3390/toxins12010025

**Published:** 2020-01-01

**Authors:** Hamed K. Abbas, Nacer Bellaloui, Alemah M. Butler, Justin L. Nelson, Mohamed Abou-Karam, W. Thomas Shier

**Affiliations:** 1Biological Control of Pests Research Unit, US Department of Agriculture-Agricultural Research Service, Stoneville, MS 38776, USA; Alemah.Butler@sanofi.com; 2Crop Genetics Research Unit, US Department of Agriculture-Agricultural Research Service, Stoneville, MS 38776, USA; nacer.bellaloui@usda.gov; 3Department of Medicinal Chemistry, College of Pharmacy, University of Minnesota, Minneapolis, MN 55455, USA; nels6685@umn.edu (J.L.N.); m_aboukaram@yahoo.com (M.A.-K.)

**Keywords:** botryodiplodin, root infection mechanism, root toxicity, *Macrophomina phaseolina*, hydroponic culture

## Abstract

Toxins have been proposed to facilitate fungal root infection by creating regions of readily-penetrated necrotic tissue when applied externally to intact roots. Isolates of the charcoal rot disease fungus, *Macrophomina phaseolina*, from soybean plants in Mississippi produced a phytotoxic toxin, (−)-botryodiplodin, but no detectable phaseolinone, a toxin previously proposed to play a role in the root infection mechanism. This study was undertaken to determine if (−)-botryodiplodin induces toxic responses of the types that could facilitate root infection. (±)-Botryodiplodin prepared by chemical synthesis caused phytotoxic effects identical to those observed with (−)-botryodiplodin preparations from *M. phaseolina* culture filtrates, consistent with fungus-induced phytotoxicity being due to (−)-botryodiplodin, not phaseolinone or other unknown impurities. Soybean leaf disc cultures of Saline cultivar were more susceptible to (±)-botryodiplodin phytotoxicity than were cultures of two charcoal rot-resistant genotypes, DS97-84-1 and DT97-4290. (±)-Botryodiplodin caused similar phytotoxicity in actively growing duckweed (*Lemna pausicostata*) plantlet cultures, but at much lower concentrations. In soybean seedlings growing in hydroponic culture, (±)-botryodiplodin added to culture medium inhibited lateral and tap root growth, and caused loss of root caps and normal root tip cellular structure. Thus, botryodiplodin applied externally to undisturbed soybean roots induced phytotoxic responses of types expected to facilitate fungal root infection.

## 1. Introduction

Charcoal rot is a plant disease caused by the fungus, *Macrophomina phaseolina* (Tassi) Goid [1], in over 500 commercially-important plant species ranging from ornamental plants to trees to major food and fiber crops, including soybean (*Glycine max* L. (Merr.)). An example of the impact of charcoal rot disease on agriculture was provided by attempts to establish commercial natural rubber production with guayule (*Parthenium argentatum* Gray) in the arid southwest region of the US as an alternative to imported material from the rubber tree (*Hevea brasiliensis*) [2]. Guayule rubber production was only competitive when plants were grown close enough together that the roots interdigitated, under which conditions charcoal rot could spread from plant to plant destroying the crop [3]. Because charcoal rot is favored by hot, dry conditions [4], it is a climate-impacted plant disease that is predicted to be an increasingly important agronomic problem going forward, given that climate change is predicted to result in hotter, drier conditions in the majority of the world [5].

Research on (−)-botryodiplodin as a food contaminant has mainly focused on its production by the blue cheese fungus, *Penicillium roqueforti* [6,7]. (−)-Botryodiplodin production by *P. paneum* in bread and silage is also a concern [8,9]. Concerns about (−)-botryodiplodin as a possible contaminant in Roquefort cheese and other foods have led to extensive studies of its possible toxic effects in mammalian systems [10]. Because *M. phaseolina* is known to produce (−)-botryodiplodin and to be present in seeds as an endophyte, contamination of food items such as tofu and vegetable oil by (−)-botryodiplodin is a concern [10]. However, studies on foods and feeds impacted by charcoal rot disease have not been reported. 

Although soybean cyst nematode is the major cause of soybean yield losses most years in most parts of the US, charcoal rot has traditionally been the most economically-important disease of soybean in the mid-southern region of the US (i.e., in Arkansas, Mississippi, and Louisiana) [11,12,13,14,15]. However, rising average temperatures and increased prevalence of drought have made the disease an increasingly important cause of yield losses during hot, dry growing seasons in all but northern parts of the US and other parts of the world [16]. Extensive studies have been carried out attempting to use selective breeding to develop soybean genotypes that are resistant to charcoal rot, but this approach has yielded only tolerant or moderately resistant genotypes [17,18]. Attempts to use various agronomic techniques to prevent the disease have also failed, so research on charcoal rot continues [17,19].

The mechanism used by *M. phaseolina* to infect soybean plants from the soil reservoir is poorly understood. *M. phaseolina* enters plants through the roots, then spreads through conductive tissues, reducing conduction volume, plant weight, and height, as well as reducing seed quality and quantity [16,20]. Inside plant tissues, *M. phaseolina* produces microsclerotia that appear as gray to black dots in and on stems and leaves and serve as reproductive structures that survive over winter in soil or as endophytes in the infested seed [21]. Fungi are widely believed to gain admission to plant roots from the soil by either (i) physical penetration of tissue; or (ii) secretion of toxins that kill plant tissue locally, creating a necrotic region through which fungal hyphae can easily propagate [19,22,23,24]. The mechanism(s) used by toxins to create localized necrosis in plant roots is not well understood, but two possible mechanisms are (i) secretion of hydrolytic enzymes or toxins that induce activation of endogenous hydrolytic enzymes; and (ii) secretion of toxins that specifically kill dividing meristematic cells near root tips, which creates necrotic tissue in a place that provides convenient access to the plant’s vascular system through which the fungus can spread throughout the plant [19]. 

*M. phaseolina* has been reported to produce several mycotoxins that are candidates for toxin-mediated initiation of infection by generating a necrotic zone. These mycotoxins include phaseolinone [25], botryodiplodin [26], and patulin, because the *M. phaseolina* genome contains genes for its biosynthetic enzymes [27]. Siddiqui et al. (1979) [25] identified phaseolinone in culture extracts of pathogenic *M. phaseolina* isolated as an endophyte of mung bean. Dhar et al. (1982) [28] proposed the structure of the isolated toxin to be an epoxidized analog of a known phytotoxin, phomenone, which is part of an extensive family of phytotoxic eremophilane sesquiterpenoid (C-15) toxins produced by numerous plant pathogenic fungi [29]. Phaseolinone has been synthesized by Kitahara et al. (1991) [30] by conversion of another known eremophilane sesquiterpenoid toxin, phomenone. A series of 12 eremophilane analogs, including synthetic phaseolinone, were shown to be phytotoxic, producing either green islands on monocot leaves or necrotic lesions on dicots [29,31]. 

Ramezani et al. (2007) [32] and Abbas et al. (2019) [33] found no detectable phaseolinone in culture extracts of *M. phaseolina* isolated from infected soybean plants in the Mississippi Delta region of the southern USA. Bioassay-guided fractionation of the extracts led to the isolation of a different, known mycotoxin, botryodiplodin, which was first isolated by Sen Gupta et al. (1966) [26] from culture filtrates of *Botryodiplodia theobromae* Pat. (syn. *Lasiodiplodia theobromae* (Pat.) Griffon & Maubl), a cellulolytic fungus first isolated in 1944 from mildewed tent fabric in India, and subsequently shown to be a plant pathogen in many economically-important crops in the tropics and sub-tropics around the world [34]. 

The objectives of the present study were to investigate the identity of the phytotoxin produced by *M. phaseolina* isolates from Mississippi soybeans with charcoal rot disease as botryodiplodin, and to characterize some botryodiplodin root toxicity properties that could enable it to play a role in the initial stages of the soybean root infection mechanism of *M. phaseolina*.

## 2. Results and Discussion

### 2.1. Synthesis of (±)-Botryodiplodin

Chemically synthesized (±)-botryodiplodin exhibited potent phytotoxicity in each of a series of experimental systems, including *L. pausicostata* axenic cultures (Figure 1), soybean leaf discs in culture (Figure 2 and Figure 3), and soybean seedling roots in hydroponic (Figure 4) and sand (Figure 5) culture. One explanation for the observation [32] that bioassay-guided fractionation of phytotoxicity produced by *M. phaseolina* isolates that cause charcoal rot disease in Mississippi soybeans yielded (-)-botryodiplodin, but no detectable phaseolinone, was that the toxin preparations contained a small percentage of either phaseolinone or another unknown, but the potent toxin that was responsible for the observed phytotoxicity. Phaseolinone was proposed by Siddiqui et al. (1979) [25] to mediate infection in charcoal rot disease based on its isolation from culture filtrates of an *M. phaseolina* endophyte from mung beans in India. When a phytotoxin is purified from nature, it is never 100% pure, so that it is always possible that the phytotoxicity may actually reside in a highly toxic impurity, rather than in the major component of the preparation. Chemical synthesis is one approach that can provide evidence that the major component of the preparation is the actual toxin. Chemical synthesis of a toxin is unlikely to produce the same impurities as found in material purified from nature. Even if the impurities in the two types of preparations are both toxic, they are unlikely to induce identical pathology in all toxicity tests. Therefore, identical phytotoxic properties are unlikely to be observed in synthetic and natural preparations of a toxin, if the activities of either are due to a highly active impurity. At least seven syntheses of botryodiplodin have been reported, since the initial success by McCurry & Abe (1973) [35]. None of these syntheses could conceivably produce phaseolinone or any other eremophilane sesquiterpenoid as a by-product. Although the method used in this study to synthesize (±)-botryodiplodin was selected because it involved only five steps using simple, standard chemistry and low cost reagents, it also could not conceivably produce phaseolinone or any other eremophilane sesquiterpenoid as a contaminant. Antibacterial activity was the first biological activity identified for (-)-botryodiplodin [26], and the easiest to assay. Chemically synthesized (±)-botryodiplodin was shown to exhibit antibacterial activity indistinguishable from that of (−)-botryodiplodin purified from *M. phaseolina* cultures [32] (data not shown). In addition, (±)-botryodiplodin induced phytotoxic responses indistinguishable from those induced by (−)-botryodiplodin, when compared in duckweed (*L. pausicostata*) plantlet cultures and soybean leaf discs in culture (see below). Identical activity of (±)-botryodiplodin and (−)-botryodiplodin is consistent with extensive studies on the mechanism of action of (−)-botryodiplodin by Moule et al. (1981a; 1981b; 1982) [36,37,38], which indicated that the toxin acts by chemical reactions in cell nuclei that covalently cross-link proteins to DNA, and not by interacting with a chiral binding site on any enzyme or receptor that might require an optically active form. Although (+)-botryodiplodin has been prepared by chemical synthesis [39], its biological activity, or the lack thereof, has not been reported by these investigators or others. More extensive structural alterations of botryodiplodin in the form of epimers have been reported to be inactive in the case of 4-*epi*-botryodiplodin [40]. Félix et al. (2019) [41] observed that cytotoxicity of 3-*epi*-botryodiplodin measured in Vero monkey kidney cells and 3T3 mouse fibroblast cultures was 0–5% of the cytotoxicity of botryodiplodin. However, in a leaf puncture assay in young tomato plant leaves, 3-*epi*-botryodiplodin produced a much larger lesion with different morphology than botryodiplodin, but similar to the lesion produced by botryodiplodin acetate. Thus, the possibility that (+)-botryodiplodin might be an inactive diluent in the (±)-botryodiplodin preparations used in this study, cannot be rigorously excluded, but if it were inactive, all conclusions drawn would be the same, with reported (−)-botryodiplodin activities occurring at half the stated concentrations. 

The simplest explanation for differences in the type of toxin produced in culture by endophytic *M. phaseolina* isolated from mung beans in India [25] and pathogenic *M. phaseolina* isolated from soybeans in Mississippi [32] is that the isolate studied by Siddiqui et al. (1979) [25] produced both phaseolinone and (−)-botryodiplodin, whereas only (−)-botryodiplodin was produced by the Mississippi isolates [10]. Production of multiple, structurally dissimilar mycotoxins by a single fungus has been well-documented [42], and there are numerous examples in the scientific literature of regional variations in mycotoxin production by the same species of fungus [43,44]. 

### 2.2. Phytotoxicity of (±)-Botryodiplodin in Lemna Pausicostata (Duckweed) Cultures

A series of studies were initiated to determine if botryodiplodin possesses properties useful for a mycotoxin to play a role in mediating root infection by *M. phaseolina* from a soil reservoir. Specifically, to be an effective mediator of root infection, a toxin must be able to kill undisturbed, actively growing root tissue in the absence of an insect, nematode, or other vector that physically damages root tissue. Root cells killed by the toxin should create a necrotic region, preferably one that would provide fungal hyphae with facile access to the plant vascular system. A toxin-mediated fungal root infection mechanism should be able to facilitate tissue entry in the absence of fungal structures such as appressoria [45] that enable fungal cells to physically penetrate plant leaf tissue in the absence of a vector. 

(±)-Botryodiplodin (0 to 64 μg/mL) dissolved in the culture medium of parallel axenic cultures of the aquatic plant of *Lemna pausicostata* (duckweed) induced a phytotoxic response in intact, growing plantlets floating on the surface of the culture medium over a 96-hour period (Figure 1) that was indistinguishable from the phytotoxic response to (−)-botryodiplodin prepared as described by Ramezani et al. (2007) [32]. Phytotoxicity was measured as percent growth reduction measured by the number of plantlet fronds produced relative to parallel control cultures not treated with (±)-botryodiplodin. Additional phytotoxicity occurred as a formation of necrotic tissue with light brownish color around the edges of the fronds and some bleaching progressing to 100% growth inhibition, 100% mortality, and complete bleaching. No detectable toxicity was observed at 24 hours, because growth was measured as frond number and more time than that was needed for a plantlet to generate a new frond under conditions used. However, the full extent of toxicity was observed at 48 hours with IC_50_ = 0.22 μg/mL. The dose-response curves at 72 hours (IC_50_ = 0.19 μg/mL) and 96 hours (IC_50_ = 0.18 μg/mL) were not significantly different from each other (Pearson’s *r* = 0.993, *p* = 0.601, multiple linear regression analysis), or from that at 48 hours (Pearson’s *r* = 0.994, *p* = 0.995 at 72 hr; r = 0.984, *p* = 0.874 at 96 hrs, multiple linear regression analysis). 

Phytotoxicity of (±)-botryodiplodin was determined in leaf discs from charcoal rot tolerant and susceptible soybean genotypes. (±)-Botryodiplodin (0 to 320 μg/mL) in culture medium for 96 hours induced the same phytotoxic response in soybean leaf discs cut from mature leaves of three- to four-week old soybean seedlings as observed with (-)-botryodiplodin [32], specifically progressive browning (necrosis) around the edges of the leaf disc and bleaching (light-induced loss of chlorophyll) progressing to complete browning of the leaf disc and 100% bleaching. (±)-Botryodiplodin phytotoxicity was compared in leaf discs from the following three genotypes: DT97-4290, which was released as a charcoal rot disease resistant soybean genotype; and two others that are considered susceptible to charcoal rot disease, DS97-84-1 and Saline. The percent severity of phytotoxic responses was quantitated at 24, 48, 72, and 96 hours using the rating scale given in Figure 2. At each time period, Saline was significantly (*p* < 0.05, multiple regression) more susceptible to the phytotoxic effects of (±)-botryodiplodin than DS97-84-1 and DT97-4290. While DS97-84-1 was more susceptible to (±)-botryodiplodin than DT97-4290 at some times, the differences were not significant. At 24 hours (Figure 3A), phytotoxicity was observed only at the highest (±)-botryodiplodin concentrations with IC_50_ values of 320 μg/mL for each of Saline, DS97-84-1, and DT97-4290, respectively. At 48 hours, (Figure 3B) phytotoxicity was observed at lower (±)-botryodiplodin concentrations with IC_50_ values of 136 μg/mL for Saline and 272 μg/mL for DS97-84-1 and DT97-4290. At 72 hours (Figure 3C), substantial phytotoxicity was observed at progressively lower (±)-botryodiplodin concentrations with IC_50_ values of 59.5 μg/mL for Saline and 132 μg/mL for DS97-84-1 and DT97-4290. At 96 hours (Figure 3D), substantial phytotoxicity was observed at much lower (±)-botryodiplodin concentrations with IC_50_ values of 14.9, 38.5, and 42.9 μg/mL for Saline, DS97-84-1 and DT97-4290, respectively. The observation that the three soybean genotypes examined in the study exhibited susceptibility to the phytotoxic effects of (±)-botryodiplodin in the order Saline > DS97-84-1 > DT97-4290 is consistent with the charcoal rot tolerance reported for genotype DT97-4290 [18] resulting from a change expressed in multiple tissues, including leaf tissue. Given that the level of resistance expressed by genotype DT97-4290 is not sufficient to prevent charcoal rot disease and infection by *M. phaseolina* [18], subsequent studies focused on investigating root-specific responses believed to be associated with initial infection. 

### 2.3. Root Toxicity of (±)-Botryodiplodin in Soybean Seedlings

Studies on root toxicity of (±)-botryodiplodin used soybean seedlings in hydroponic culture with the toxin being added to culture medium bathing only the roots. Soybean seedlings in hydroponic culture were treated for four days with a range of (±)-botryodiplodin concentrations (10 to 80 μg/mL) in the nutrient solution bathing the roots. Control seedlings produced abundant lateral roots during the hydroponic culture period. The addition of (±)-botryodiplodin to the nutrient solution reduced lateral root production even at 10 μg/mL, the lowest concentration tested in initial trials (Figure 4). Inhibition of root growth by (±)-botryodiplodin treatment was quantified by the dry weight relative to that of control plants exposed to 0 μg/mL (±)-botryodiplodin. There was significantly greater toxicity to lateral roots than to tap roots (*p* < 0.05, regression analysis) (Figure 6). (±)-Botryodiplodin exposure resulted in about an eight-fold reduction in lateral root growth, but only in about a two-fold reduction in tap root growth. There was significant reduction in tap root growth at the highest (±)-botryodiplodin concentrations tested (≥40 μg/mL), but the IC_50_ (23.5 μg/mL) was 5.6-fold higher than the IC_50_ for lateral roots (4.2 μg/mL), which exhibited significant (*p* < 0.05, Student’s *t*-test) reduction in lateral root growth at ≥5 μg/mL (Figure 6). Thus, botryodiplodin caused toxicity to undisturbed soybean roots when applied externally, which is a property expected for a toxin capable of playing a role in facilitating fungal root infection from a soil reservoir (Figure 4 and Figure 6). 

A set of experiments exposing soybean seedling roots to (±)-botryodiplodin in sand culture was conducted to eliminate the possibility that physical contact of soybean roots with solid soil particles might induce or maintain a protective layer on roots. Seedlings germinated in soil were transplanted to sand culture, acclimatized, and then the roots exposed to (±)-botryodiplodin (10, 100, and 300 μg/mL) dissolved in fresh culture medium (Figure 5). (±)-Botryodiplodin treatment resulted in greatly reduced lateral root production, particularly at the higher concentrations. There was no indication that the sand used in the sand culture system interfered with phytotoxicity either by adsorption of toxin on silica surfaces, or by interfering with conduct of the experiment either by preventing continuous visual monitoring of toxin-induced damage or causing root damage when washing sand away. 

Pink to red discoloration of exposed roots occurred at the highest (±)-botryodiplodin concentration (300 μg/mL) in sand culture (Figure 5), and at the higher concentrations tested in liquid hydroponic culture (Figure 4), with the darkest coloration at the highest concentration (80 μg/mL). Formation of pigment by reaction of botryodiplodin with protein and other amines has been observed numerous times [26,35,39,46,47]. The (±)-botryodiplodin used in the present study has been shown to react with proteins, amino acids, and a wide variety of other amines to give red to yellow pigments [48]. Given that soybean seedlings have been reported to express proteins such as nutrient and water transporters on root surfaces [49], the pink to red pigment observed on soybean seedling roots treated with (±)-botryodiplodin in hydroponic culture (Figure 4 and Figure 5) may have formed by a similar reaction with root surface proteins.

The production of abundant lateral roots by soybean seedlings under the stationary hydroponic conditions used in this study (Figure 4 and Figure 6) presumably results from disruption of oxygen and ethylene exposure to roots, which has been shown in *Arabidopsis thaliana* to be genetically defined and environmentally regulated [50,51]. Soybean has similar ethylene receptors and associated regulatory gene products [52], which provides an explanation for the well-documented occurrence of lateral root production by soybean when soil becomes waterlogged [53,54]. In plant root growth, cell division occurs solely in meristematic regions near root caps, and root extension primarily results from subsequent cell elongation. Botryodiplodin has been shown to target DNA synthesis and dividing cells in a wide variety of biological systems, including bacteria [26,55], fungi [56], yeast [26], plants [32], and mammalian cells [35,37,57,58,59]. A phytotoxin such as (−)-botryodiplodin, which kills dividing cells, would be expected to target meristematic tissue near the tips of both tap and lateral roots. The higher reduction in lateral root growth (~eight-fold) than in tap root growth (~two-fold) by (±)-botryodiplodin (Figure 6) is consistent with the toxin acting on meristematic tissue, which makes up a larger percentage of total tissue weight in small lateral roots than it does in the larger tap root.

Soybean seedlings growing in hydroponic culture with roots exposed to (±)-botryodiplodin (15 µg/mL) in culture medium (Figure 7) resulted in the loss of the root cap and meristematic tissue without involvement of a vector or physical injury, and were consistent with the toxin targeting dividing cells in the meristem. Similar loss of the root cap and meristematic tissue occurred at the higher (±)-botryodiplodin concentrations tested (35 and 80 µg/mL). Additional studies are needed to determine how rapidly the root tip loss occurs at various (±)-botryodiplodin concentrations. 

Thus, (±)-botryodiplodin applied externally to undisturbed soybean roots induced phytotoxic responses of a type expected to facilitate fungal root infection. An example of a plausible root infection mechanism involving the observed responses of soybean root to (±)-botryodiplodin could involve hyphae of a fungus like *M. phaseolina* propagating outward from a plant-derived nutrient source through the soil in all directions until hyphae detect the presence of a root tip, stimulating release of (−)-botryodiplodin. The released (−)-botryodiplodin would be expected to cause loss of the root tip and exposure of the vascular system that should facilitate the propagation of fungal hyphae into the vascular system and subsequently throughout the plant [19]. However, additional studies will be needed to confirm that targeting of root tip meristematic cells is involved in the actual root infection mechanism used by *M. phaseolina* in charcoal rot disease of soybeans in the field. 

## 3. Conclusions

The toxin, botryodiplodin, produced by *M. phaseolina*, the fungus that causes charcoal rot disease in many plant species, is phytotoxic in soybean leaf disc cultures and in actively growing *Lemna pausicostata* plantlet cultures. Botryodiplodin exposed to undisturbed roots of soybean seedlings in hydroponic culture results in a root tip destruction response that would facilitate fungal infection of the root. 

## 4. Materials and Methods

### 4.1. Preparation of (±)-Botryodiplodin

(±)-Botryodiplodin was selected for use in these studies, because it is readily synthesized chemically in larger amounts than were available by fermentation [32]. The mechanism of action of botryodiplodin has been extensively studied by Moule et al. [36,37,38], who provided evidence for non-enzymatic (i.e., chemical) crosslinking of DNA to protein. There have been no reports of botryodiplodin binding specifically to a chiral binding site on any enzyme or receptor. A non-enzymatic mechanism of action for botryodiplodin would result in phytotoxicity of synthetic (±)-botryodiplodin being equivalent to that of fermentation-derived (−)-botryodiplodin. The (±)-botryodiplodin used in this study was synthesized by preparing α-methyl-α-angelicalactone, using a modification of the method of Helberger et al. (1949) [60], followed by its conversion to the final product using four steps that are included in the synthetic method developed by Mukaiyama et al. (1974) [61] (Figure 8). Briefly, α-methyllevulinic acid (**1**) (500 mg) (TCI America, Portland, OR, USA), was treated with phosphoric acid (1% wt/wt) and subjected to vacuum distillation at 120–130 °C and ~40 Torr to provide α-methyl-α-angelicalactone (**2**) in approximately 80% yield. The product was treated under argon with boron trifluoride etherate and formaldehyde generated in situ by thermal degradation of paraformaldehyde. The reaction was quenched with NaHCO_3_ aqueous solution and extracted into dichloromethane. The product, (±)-cis-α-methyl-β-acetyl-γ-butyrolactone (**3**), was purified by chromatography on silica gel in diethyl ether:hexane 4:1 and crystallized from hexane. The ketone group of **3** was blocked with ethanethiol in the presence of zinc chloride and the product **4** extracted into dichloromethane and purified by chromatography on silica gel in diethyl ether:hexane 4:1. Reduction of **4** with diisobutylaluminium hydride in tetrahydrofuran at −78 °C yielded the diethanethiol derivative of (±)-botryodiplodin (**5**), which was purified by chromatography on silica gel using a 5% to 20% diethyl ether:hexane gradient. Unblocking of lactol **5** in acetone containing 1% water, CuCl_2_ and CuO was accomplished at room temperature in 30–60 minutes. (±)-Botryodiplodin (**6**) was extracted from the reaction mixture into dichloromethane and purified by chromatography on silica gel using ether:hexane 4:1 followed by re-chromatography on silica gel using dichloromethane:methanol 20:1 to yield 116 mg (20.9% overall yield) at a purity of >98% based on thin layer chromatography and nuclear magnetic resonance spectroscopy. The (±)-botryodiplodin (**6**) exhibited ^1^H nuclear magnetic resonance spectroscopy values and thin layer chromatographic R_f_ values identical to those reported in the literature [39,62] and those obtained in this laboratory with (−)-botryodiplodin purified from cultures of *M. phaseolina* [32], except that (±)-botryodiplodin was not optically active. 

### 4.2. Assay of Antibacterial Activity of Botryodiplodin

The biological activity level of (±)-botryodiplodin was confirmed using antibacterial activity, the type of activity used to guide the initial isolation of (−)-botryodiplodin by Sen Gupta et al. (1966) [26], and the most easily measured of its numerous reported biological activities, including antifungal [56], phytotoxic [25,32], anti-cancer [37,57], mutagenic [36], and antifertility [58] activities. Antibacterial activity was compared on samples of (±)-botryodiplodin, prepared as described above, and (−)-botryodiplodin purified as described by Ramezani et al. (2007) [32] from culture filtrates of *M. phaseolina* isolated from a soybean plant with charcoal rot disease in Mississippi. Antibacterial activity was measured by serial dilution from 20 to 0.1 µg/mL in Mueller-Hinton broth in triplicate in the wells of a 96-well tray using (±)-botryodiplodin and (−)-botryodiplodin samples sterilized by dissolution at 10 mg/mL in 95% ethanol. The wells were inoculated with an actively growing culture of *Bacillus subtilis*, strain 1a1, isolated in this laboratory from lawn soil and shown to be susceptible to all antibiotics in a 28-member panel except thiostrepton. Trays were cultured overnight at 37 °C and bacterial growth estimated as the OD at 600 nm in a plate reader (BioTek Instruments Synergy HT, Winooski, VT, USA). 

### 4.3. Plant Growth and Environmental Conditions

Soybean genotypes DT97-4290 (moderately resistant to charcoal rot) [18], DS97-84-1 and Saline [63] (both of which are susceptible to charcoal rot) were grown in the greenhouse. Seeds were planted and germinated in flat trays of vermiculite, and similarly sized seedlings were transplanted into 9.45 L pots filled with a silt loam soil (24% sand, 54% silt, and 22% clay with 1.3% organic matter) pH 6.5, 17 cmol/kg cation exchange capacity. Plants were watered as needed to maintain soil water potential at field capacity, i.e., between –15 to –20 kPa. Four pots were used for each soybean genotype, and three plants were grown in each pot. Greenhouse conditions were about 34 °C ± 8 °C during the day and approximately 28 °C ± 6 °C at night with a photosynthetic photon flux density of about 850–2100 μmol·m^−2^ s^−1^, as measured by Quantum Meter (Spectrum Technologies, Inc., Aurora, IL, USA). The range of light intensity reflects the range from a cloudy day (850 μmol·m^−2^ s^−1^) to a sunny day (2100 μmol·m^−2^ s^−1^). The source of lighting in the greenhouse was a mixture of natural and artificial lights. Plant leaves were harvested during the vegetative phase of growth. 

### 4.4. Phytotoxicity of (±)-Botryodiplodin in Soybean Leaf Discs

Dose-response curves were obtained for phytotoxic responses to a range of (±)-botryodiplodin concentrations by triplicate cultures of three soybean leaf discs. Leaf discs cut from healthy leaflets from true mature leaves of 3- to 4-week-old plants of each of the three soybean types were used to determine the phytotoxicity of (±)-botryodiplodin. All leaves were harvested in the laboratory and three soybean leaf discs measuring 4-mm diameter were cut with a sterile cork borer (#4) and placed in sterile 24-well tissue culture plates with low evaporative lids (Becton Dickinson and Company, Franklin Lakes, NJ). (±)-Botryodiplodin solutions in water (1.5 mL) over a range of concentrations (0, 2.5, 5.0, 10, 20, 40, 80, 160, and 320 µg/mL) were added to the wells of plates in triplicate. Leaf discs were incubated in a growth chamber at 25 °C under continuous visible light for 96 h and examined for signs of phytotoxicity after 24, 48, 72, and 96 h using the following symptom rating scale: Healthy tissue, 0%; a narrow zone of brown (necrotic) tissue forming around the edges of the leaf disc, 10%; a substantial zone of brown tissue forming around the edges of the leaf disc, 25%; brown tissue throughout the leaf disc, 50%; brown tissue throughout the leaf disc with bleaching, 75%; complete bleaching of the leaf disc, 100% (Figure 2). 

### 4.5. Phytotoxicity of (±)-Botryodiplodin in Duckweed Plant Cultures

Dose-response curves were obtained for phytotoxic responses to a range of (±)-botryodiplodin concentrations by triplicate cultures of three-frond duckweed plantlets. Cultures containing three duckweed (*Lemna pausicostata* Helgelm.) plantlets were used to bioassay phytotoxicity, as described by Tanaka et al. (1993) [64], with some modification. Briefly, three duckweed plantlets containing three fronds each were transferred from a laboratory maintenance culture with clean forceps to each well of a sterile 24-well tissue culture plate with a low evaporation lid. Aliquots (1.5 mL) of culture medium containing a range of (±)-botryodiplodin concentrations (0, 0.03, 0.06, 0.13, 0.25, 0.5, 1, 2, 4, 8, 16, 32, and 64 µg/mL) were added in triplicate to the wells of culture plates. Duckweed plants were subsequently incubated in a growth chamber at 25 °C under continuous light for 96 hr. Duckweed plantlets were observed for signs of phytotoxicity after 24, 48, 72, and 96 hr. Growth was measured as addition of fronds in treated cultures relative to control cultures not treated with (±)-botryodiplodin. No additional fronds being produced in a treated culture was scored as 100% inhibition of growth. 

### 4.6. Hydroponic Culture of Soybean Seedlings

The effects of (±)-botryodiplodin on soybean root growth were investigated in soybean seedlings germinated from a commercial soybean seed variety assumed to be charcoal rot-susceptible (Kansas Soybean Commission, Topeka, KS, USA) in autoclaved soil and grown under continuous light to the cotyledon stage (VC, 4–7 cm). Seedling roots were washed free of soil particles and transplanted to hydroponic growth medium. Seedlings were grown under hydroponic conditions for four days before use in root toxicity assays in individual 16 × 100 mm glass tubes containing 5 mL of a mixture of 90% distilled water and 10% Villagarcia medium [65]. The Villagarcia medium used consisted of distilled water (999 mL) containing CaSO_4_.2H_2_O (690 mg), KH_2_PO_4_ (34 mg), KNO_3_ (200 mg), MgSO_4_.7H_2_O (61 mg), and 1 mL of 1000-fold concentrated microsolute nutrient solution containing FeSO_4_.7H_2_O (50 mg), KCl (14 mg), H_3_BO_4_ (5.7 mg), MnSO_4_.H_2_O (1.5 mg), ZnSO_4_.7H_2_O (2.6 mg), CuSO_4_.5H_2_O (0.45 mg), and (NH_4_)_6_Mo_7_O_24_ (2.1 mg). Seedlings were held in place by the tube walls and maintained with roots covered with medium added daily as needed. Seedlings placed under sand culture conditions were grown four days in 5 mL of washed, sterile sand, which was added after the seedling was placed in the tube and kept soaked with 10% (*v*/*v*) Villagarcia medium in water added daily as needed. 

### 4.7. Root Toxicity of (±)-Botryodiplodin in Soybean Seedlings in Hydroponic Culture

Dose-response curves were obtained for phytotoxic responses to a range of (±)-botryodiplodin concentrations by the roots of groups of three soybean seedlings cultured individually in hydroponic medium. Soybean seedlings were grown in continuous light for four days at room temperature in 5 mL of hydroponic growth medium consisting of 10% Villagarcia medium and 90% water in individual 16 × 100 mm glass tubes, using the walls of the glass tubes to hold the seedlings upright. The medium was withdrawn from seedling cultures with a Pasteur pipet, replaced by fresh medium containing a range of (±)-botryodiplodin concentrations in triplicate in three individual glass tubes (0 to 300 μg/mL in an initial range-finding assay, and 0 to 80 μg/mL in subsequent studies), and cultured for an additional four days at room temperature in continuous light. Root growth was quantified by removing seedlings from the culture tubes and washing the roots with a stream of deionized water from a wash bottle. Roots were excised at the stem line with a scalpel. Lateral roots were cut from the tap roots and the two root types dried separately overnight under vacuum in a desiccator over Drierite desiccant at room temperature. Lateral and tap roots were weighed separately on a sensitive balance (Mettler Toledo UMX2 Ultra-Microbalance, Mettler-Toledo International, Columbus, Ohio), and the dry weights of triplicate samples plotted as mean ±standard error versus (±)-botryodiplodin concentration. 

### 4.8. Light Micrographs of Soybean Seedling Roots Exposed to (±)-Botryodiplodin in Hydroponic Culture

Soybean (commercial variety) seedlings were established in hydroponic culture as described above, then transplanted to individual new 16 × 100 mm glass tube containing 5 ml of (±)-botryodiplodin (0, 15, 35, and 80 μg/ml) in 10% Villagarcia medium and 90% water. Seedlings were incubated at room temperature for 4 days with continuous light, at which time control seedlings in 0 μg/ml (±)-botryodiplodin had abundant lateral roots. Seedlings in 15 μg/ml (±)-botryodiplodin had substantially reduced numbers of lateral roots and seedlings in 35 and 80 μg/ml (±)-botryodiplodin had stunted roots stained pink. An Xacto knife was used to cut the roots off at slightly above where the root begins. The excised roots were placed in labeled glass scintillation vials filled to the top with Karnovsky’s fixative [66]. Root samples were embedded in resin and thick-sectioned on an Ultracut UCT microtome (Leica, Buffalo Grove, IL, USA) using a diamond knife. Sections were collected on glass slides, stained with toluidine blue, and imaged using bright-field light microscopy on an Eclipse 90i (Nikon Inc., Melville, NY, USA) with a D2-Fi2 color camera running Nikon Elements software. 

### 4.9. Data Analysis

Phytotoxic responses were quantified as IC_50_ values (the concentration of (±)-botryodiplodin that causes 50% of the maximal toxic response) determined graphically by interpolation on plots of toxic response versus log (±)-botryodiplodin concentration prepared using the graphing package included in Microsoft Excel 2010. Statistical analyses (correlation analysis, multiple linear regression analysis, Student’s *t*-test) were conducted using the statistical package included in Microsoft Excel 2010. *p* ≤ 0.05 was considered significant. 

## Figures and Tables

**Figure 1 toxins-12-00025-f001:**
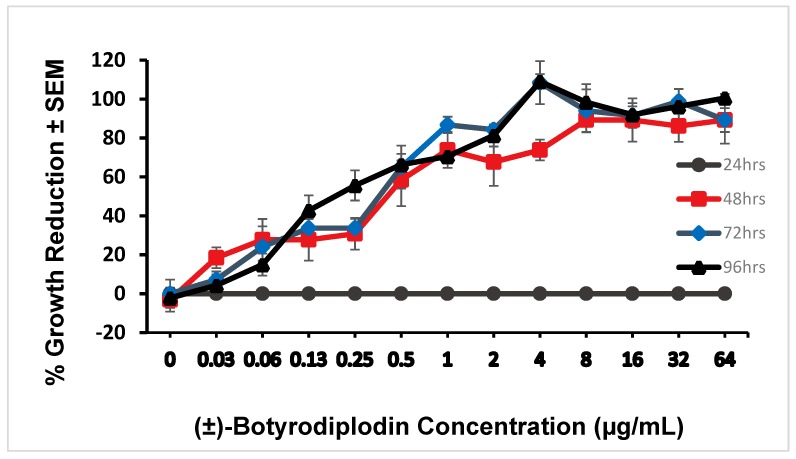
Inhibition of duckweed (*Lemna pausicostata*) plantlet growth in axenic cultures containing a range of concentrations of (±)-botryodiplodin in the culture medium. Duckweed growth was measured as percent inhibition of frond production ± SEM relative to controls not treated with toxin. Phytotoxicity was assessed at 24 h (●), 48 h (■), 72 h (♦), and 96 h (▲). The full toxic response was observed by 48 h (IC_50_ = 0.22 µg/mL); that is, the percent growth reduction at 48, 72, and 96 h were not significantly different from each other, but all were significantly greater than that at 24 hours, *p* < 0.05, multiple linear regression analysis.

**Figure 2 toxins-12-00025-f002:**
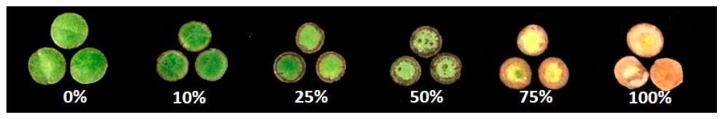
Phytotoxicity rating scale used to determine the percent severity of (±)-botryodiplodin phytotoxicity on soybean leaf discs, in which 0% = healthy tissue; 10% = slight browning around the edges of the leaf disc; 25% = moderate browning around the edges of the leaf disc; 50% = browning around the edges of the leaf disc with slight bleaching; 75% = extensive browning of the leaf disc with bleaching; and 100% = complete bleaching of the leaf disc.

**Figure 3 toxins-12-00025-f003:**
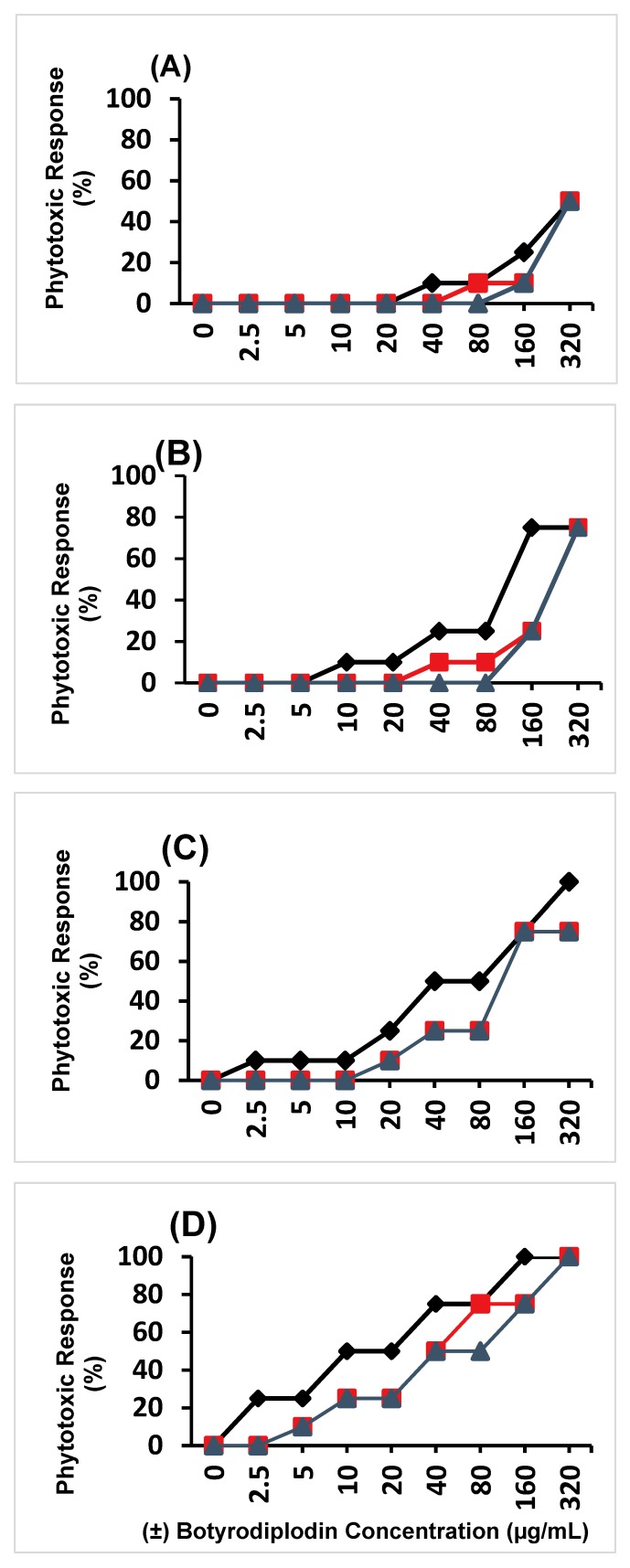
Phytotoxicity effects of (±)-botryodiplodin in cultured leaf discs from three different soybean genotypes, DT97-4290 (▲), which was released as a charcoal rot disease resistant genotype, Saline (♦) and DS97-84-1 (■). The phytotoxic response is shown at (**A**) 24 h, (**B**) 48 h, (**C**) 72 h and (**D**) 96 h. The phytotoxicity rating scale is described in Figure 2. Saline was significantly (*p* < 0.05, multiple regression) more susceptible to the phytotoxic effects of (±)-botryodiplodin than DS97-84-1 and DT97-4290 at each time point. Results are the mean of three replicates.

**Figure 4 toxins-12-00025-f004:**
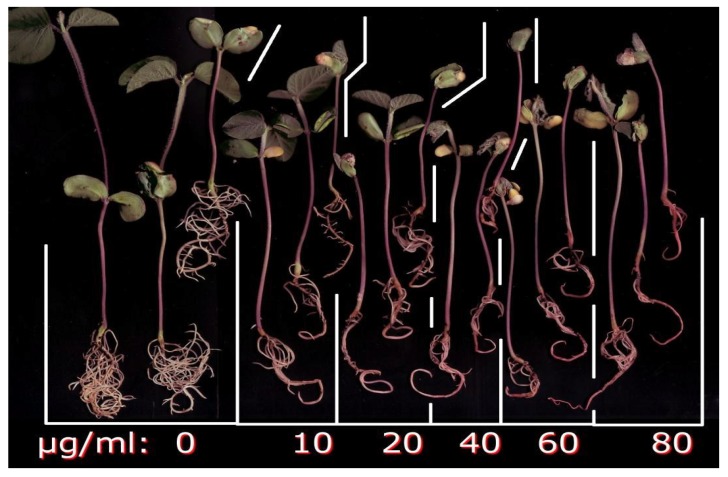
Effects of various (±)-botryodiplodin concentrations (0 to 80 µg/mL) in hydroponic culture medium on soybean seedlings. A reduced number of lateral roots and discoloration occurred at all (±)-botryodiplodin concentrations tested.

**Figure 5 toxins-12-00025-f005:**
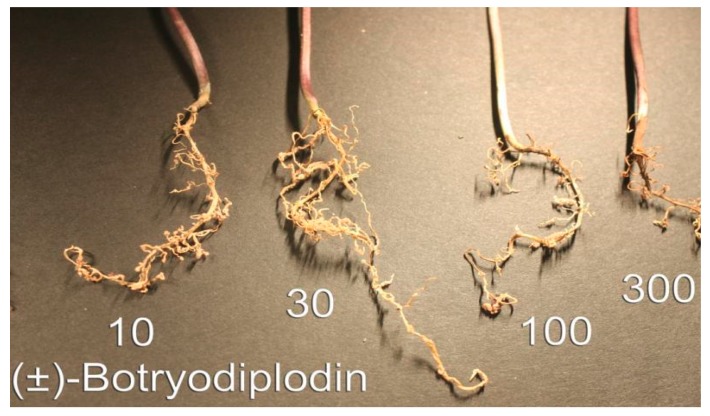
Soybean seedling roots treated with a range of concentrations of (±)-botryodiplodin (10 to 300 µg/mL) in sand culture served as controls for unsupported soybean seedling roots in hydroponic culture. A reduced number of lateral roots and discoloration occurs at higher concentrations of (±)-botryodiplodin.

**Figure 6 toxins-12-00025-f006:**
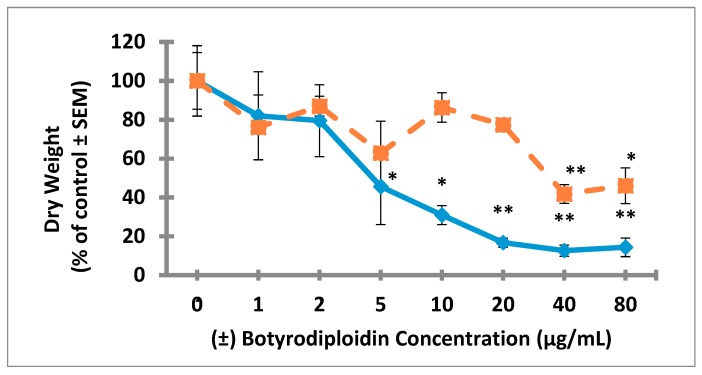
Inhibitory effects of various (±)-botryodiplodin concentrations in hydroponic culture medium (0 to 80 µg/mL) on lateral soybean seedling root growth (---◊---) with inhibition at IC_50_ = 4.2 µg/mL (100% dry weight = 11.5 ± 2.1 mg), and on tap root growth (- - - ■ - - -) with inhibition at IC_50_ = 23.5 µg/mL (100% dry weight = 13.8 ± 2.0 mg). Root growth presented on the vertical axis was measured as dry weight of excised lateral or tap roots after (±)-botryodiplodin exposure for 96 h at room temperature in continuous light. Results are the mean of three replicates ± SEM. * Significantly reduced soybean root growth at *p* < 0.05; ** significantly reduced soybean root growth at *p* < 0.01 (Student’s *t*-test).

**Figure 7 toxins-12-00025-f007:**
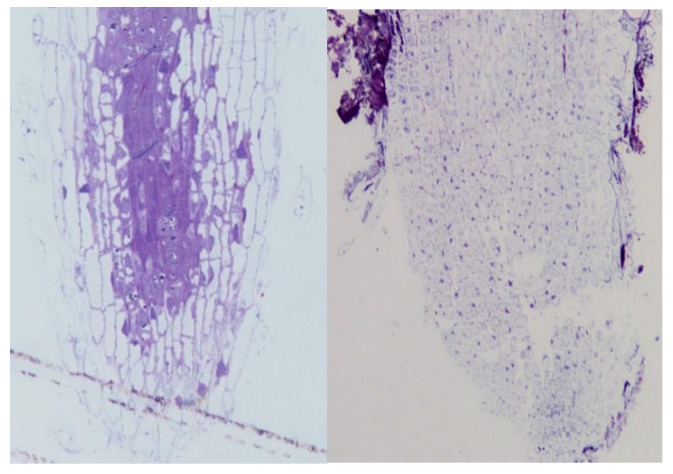
Light micrographs of root tips of soybean seedlings after four days in hydroponic culture in 10% Villagarcia medium in water with no (±)-botryodiplodin (left panel, 400×) or with (±)-botryodiplodin (15 µg/mL) (right panel, 200×).

**Figure 8 toxins-12-00025-f008:**
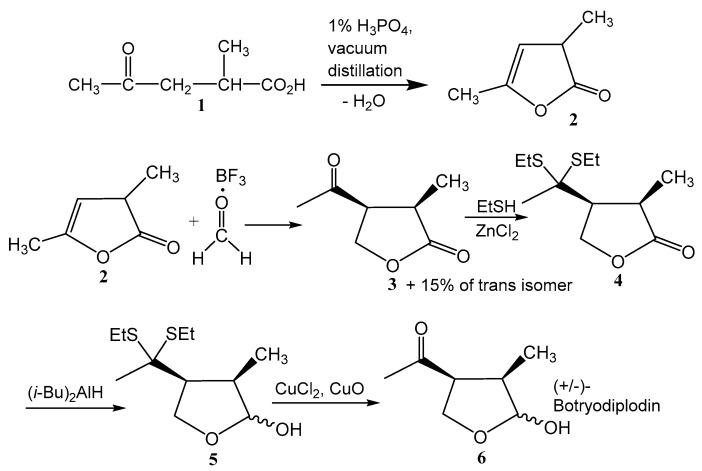
Chemical synthesis of (±)-botryodiplodin.

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
