# Peer review of "Phytotoxic Responses of Soybean (Glycine max L.) to Botryodiplodin, a Toxin Produced by the Charcoal Rot Disease Fungus, Macrophomina phaseolina"

_toxins, 2020, doi:10.3390/toxins12010025_

Round 1

Reviewer 1 Report

The manuscript entitled Botryodiplodin, a toxin produced by the charcoal rot disease fungus, Macrophomena phaseolina, induces phytotoxic responses in soybean (Glycine max L.) that are of types that could be used to facilitate root infection reported the phytotoxicity and potential role of (-)-botryodiplodin in charcoal rot disease in soybean. In this respect (±)-botryodiplodin was synthetized and assayed.

In my opinion the manuscript cannot be publish on Toxins because it showed some shortcomings and didn’t show original information.

Although some studies hypothesize a not enzymatic mechanism of toxicity of (-)-botryodiplodin, others showed differences between some its stereoisomers. In particular Fujimoto et al 1980 [Recherche toxicologique des substances métaboliques de Penicillium carneo-lutescens. Chemical and Pharmaceutical Bulletin, 28 (4), 1062-1066)] observed that the C-4 epimer, deriving from the treatment of (-)-botryodiplodin with NaHCO3, is non-toxic. More recently the natural C-3 epimer, 3-epi-botryodipldin, showed different cytotoxicity compared to (-)-botryodiplodin (Félix, C., Salvatore, M. M., DellaGreca, M., Ferreira, V., Duarte, A. S., Salvatore, F., ... & Andolfi, A. (2019). Secondary metabolites produced by grapevine strains of Lasiodiplodia theobromae grown at two different temperatures. Mycologia, 111(3), 466-476.)

There is no doubt that amount obtained from natural sources is the main problem in the biological characterization of natural compounds. Synthetic strategies can resolve this problem but the stereochemical aspects must always be considered. The direct comparison between the natural compound, in this case (-)- botryodiplodin and the racemic mixture may be the only way to have no doubts. In the abstract the authors seem to consider this (lines 12-15) but no information is reported on this through the manuscript.

For example about antibacterial activity they reported at lines 137-140 the experimental procedures but at lines 258-260 “data not show”. Also phytotoxicity between (±)- and (-)-botryodiplodins on duckweed did,’t show.

The synthesis is already reported therefore it must not be described in detail.

Author Response

Reviewer #1 states “The synthesis is already reported therefore it must not be described in detail.” As stated at ll. 252-254, original manuscript, at least eight syntheses of botryodiplodin have been reported.  However, the synthesis of (±)-botryodiplodin used in this study differs from all published syntheses, in that it combines four steps from the longer Mukaiyama et al. (1974) synthesis with one step of other, known synthetic chemistry.  Specifically, the synthesis used in this study combined a modification of the method of Helberger et al. (1949) to prepare α-methyl-α-angelicalactone, an intermediate that was converted to (±)-botryodiplodin using four steps that were taken from the method of Mukaiyama et al. (1974).  Thus, there are no novel chemical reactions that would justify publication in an organic chemistry journal, but sufficient difference from published methods for a disclosure of an outline of the method to be appropriate.  The description of the method given in the original manuscript is in considerably less detail than that which would be given in a full publication in chemistry. 

In the revised manuscript, the nature of the differences between the synthesis used in this study and published methods are made clearer in two places in the manuscript.  First, in the Materials and Methods section, the sentence (ll. 104-106, original manuscript) as follows:

The synthesis (Fig. 1) used a modification of the method of Helberger et al. (1949) to prepare the starting material, α-methyl-α-angelicalactone, which was converted to the final product by the method of Mukaiyama et al. (1974).

is replaced with

The (±)-botryodiplodin used in this study was synthesized by preparing α-methyl-α-angelicalactone, using a modification of the method of Helberger et al. (1949), followed by its conversion to the final product using four steps that are included in the synthetic method developed by Mukaiyama et al. (1974) (Fig. 1). 

Second, in the Results and Discussion section, the following sentence on ll. 255-256, original manuscript: 

The method of Mukaiyama et al., (1974) was used, because the synthesis involved only five steps using simple, standard chemistry and low cost reagents. 

has been replaced with: 

Although the method used in this study to synthesize (±)-botryodiplodin was selected because it involved only five steps using simple, standard chemistry and low cost reagents, it also could not conceivably produce phaseolinone or any other eremophilane sesquiterpenoid as a contaminant.

Reviewer #1 apparently felt the discussion of the mechanism of action of botryodiplodin and of structure-activity relationships was insufficient. Reviewer #1 cited two publications describing the lack of activity with two different epimers of botryodiplodin.  We are pleased to address this issue in two places in the manuscript.  First, we are happy to expand the introduction to the subject in the Materials and Methods section by replacing the following sentence (ll. 104-106, original manuscript): 

Given the widely accepted mechanism of action of botryodiplodin as non-enzymatic (i.e., chemical) crosslinking of DNA to protein (Moule et al., 1981a; 1981b; 1982), the phytotoxicity of synthetic (±)-botryodiplodin was expected to be indistinguishable from that of fermentation-derived (-)-botryodiplodin, as was observed (see below).

With the following: 

The mechanism of action of botryodiplodin has been extensively studied by Moule et al. (1981a; 1981b; 1982), who provided evidence for non-enzymatic (i.e., chemical) crosslinking of DNA to protein.  There have been no reports of botryodiplodin binding specifically to a chiral binding site on any enzyme or receptor.  A non-enzymatic mechanism of action for botryodiplodin would result in phytotoxicity of synthetic (±)-botryodiplodin being equivalent to that of fermentation-derived (-)-botryodiplodin. 

Second, in the Results and Discussion section 3.1, the following sentence on ll. 264-273, original manuscript: 

Identical activity of (±)-botryodiplodin and (-)-botryodiplodin is consistent with the accepted mechanism of action of (-)-botryodiplodin (Moule et al., 1981a; 1981b; 1982), specifically that the toxin acts by chemical reactions in cell nuclei that covalently cross-link proteins to DNA, and not by interacting with a chiral binding site on any enzyme or receptor that might require an optically active form. Although (+)-botryodiplodin has been prepared by chemical synthesis (Rehnberg & Magnusson, ‎1990), its biological activity, or the lack thereof, was not reported.

has been replaced with: 

Identical activity of (±)-botryodiplodin and (-)-botryodiplodin is consistent with extensive studies on the mechanism of action of (-)-botryodiplodin by Moule et al. (1981a; 1981b; 1982), which indicated that the toxin acts by chemical reactions in cell nuclei that covalently cross-link proteins to DNA, and not by interacting with a chiral binding site on any enzyme or receptor that might require an optically active form.  Although (+)-botryodiplodin has been prepared by chemical synthesis (Rehnberg & Magnusson, ‎1990), its biological activity, or the lack thereof, has not been reported by these investigators or others.  More extensive structural alterations of botryodiplodin in the form of epimers have been reported to be inactive in the case of 4-epi-botryodiplodin (Fujimoto et al., 1980) and almost inactive in the case of 3-epi-botryodiplodin (Félix et al., 2019). 

The following two references have been added: 

Félix C, Salvatore MM, DellaGreca M, Ferreira V, Duarte AS, Salvatore F, Naviglio D, Gallo M, Alves A, Esteves AC, Andolfi A. 2019. Secondary metabolites produced by grapevine strains of Lasiodiplodia theobromae grown at two different temperatures. Mycologia, 111:466-476. Fujimoto Y, Kamiya M, Tsunoda H, Ohtsubo K, Tatsuno T. 1980. Recherche toxicologique des substances métaboliques de Penicillium carneo-lutescens. Chemical and Pharmaceutical Bulletin, 28:1062-1066.

Reviewer 2 Report

This study was undertaken to determine if (-)-botryodiplodin induces toxic responses of types that could facilitate root infection. The study designe is interesting. The paper is well structured and the methods are well described. I provide some comments and suggestion to improve the manuscript:

1) Title

I suggest to short the title;

2) Introduction/Disscusion

Please complete and update the references with items from the last 5 years. Currently in references you can find just few articles from last 10 years, and many olders (since 1949), which can be not actual anymore;

3) Keywords

please add: botryodiplodin;

4) Results

If the measurements were repeated 3 times, I do not understand why the / - SD values were not indicated in the results. SD values are also not plotted on all of the charts;

5) Figure 6

more details should be provided in the description of the chart axis (y); some details are not proper (dot instead of rhombus).

Author Response

Reviewer #2 suggested that the title be shortened. In the revised version the title has been shortened to “Phytotoxic responses of soybean (Glycine max L.) to botryodiplodin, a toxin produced by the charcoal rot disease fungus, Macrophomina phaseolina”.  Reviewer #2 asked that the Introduction/Discussion include more references with items from the last 5 years. In the revised manuscript we have added the following reference to the accompanying paper on line 87, original manuscript changing “Ramezani et al. (2007) found no detectable . . “ to “Ramezani et al. (2007) and Abbas et al. (2019) found no detectable . . “:  Abbas HK, Bellaloui N, Accinelli C, Smith JR, Shier WT. 2019. Toxin production in soybean (Glycine max L.) plants with charcoal rot disease and by Macrophomina phaseolina, the fungus that causes the disease. Toxins 11:645; doi:10.3390/toxins11110645.

We are pleased to make this addition.  However, in reality, botryodiplodin is a neglected toxin, with the result there is a lack of other relevant references appearing in the last 5 years.  We hope our discoveries on the roles of botryodiplodin in root infection will change that. 

Reviewer #2 asked that the word “botryodiplodin” be added to the Keywords. That has been done on line 25, revised manuscript.  Reviewer #2 asked for clarification on the lack of statistics for results of soybean leaf disc assays presented in Figure 4. The values presented are arbitrary percentages in a process that can not be quantitated.  Thus the average of 25% and 50% is not meaningful as a statistic. Means are used because the graphs become to cluttered for easy understanding if each of the 3 points were individually plotted.    Reviewer #2 asked that more details be provided in the description of the vertical axis of Figure 6. Reviewer #2 also pointed out that a dot has been used on the legend instead of a rhombus. In the revised manuscript the legend to Figure 6 (lines 378 to 383) has been modified by adding dashed flanking lines to the solid square symbol and almost undashed flanking lines to the rhombus symbol.  Unfortunately, the Symbol package in the software available to us does not have a solid rhombus.  We would like to request the editorial staff replace the open rhombus in the revised manuscript with a solid rhombus.  In addition the sentence “Root growth presented on the vertical axis was measured as dry weight of excised lateral or tap roots after (±)-botryodiplodin exposure for 96 h at room temperature in continuous light.” was added at line 381, original manuscript. 

Reviewer 3 Report

​The manuscript deals with a set of assays aimed at demonstrating the involvement of botryodiplodin in the pathogenesis of Macrophomina phaseolina. The authors were successful in demonstrating that a chemically synthesized botryodiplodin inhibited the root growth in soybean seedlings under both hydroponic and sand culture conditions. Using light microscopy, they realized that root meristems were compromised by the exposure to the toxin. Based on these observations and the literature, the authors argued that the alterations induced by botryodiplodin may be associated with the M. phaseolina pathogenesis. In order to corroborate such a statement, a leaf-disc assay was established to test the effect of botryodiplodin on three soybean varieties differing for their susceptibility level to the pathogen. Actually, from this assay I cannot see marked differences between the resistant variety and the susceptible ones. This was explained by the authors with the fact the resistance could occur mainly in the roots rather than in the leaves. So I would expect other assays reported in the manuscript to be performed on these three varieties as well, but they were carried out with only one variety. Therefore, the phytotoxic effect of botryodiplodin was definitely demonstrated, but its involvement in the pathogenesis would need the test on both resistant and susceptible varieties. In fact, by testing only a susceptible variety, it cannot be understood whether botryodiplodin is responsible for universal phytotoxicity or, as hypothesized, for pathogenesis-related phytotoxicity.  
Therefore, I would recommend taking into account this consideration​​ and discuss it in the manuscript text. It could be necessary to avoid the emphasis on the fact that the phytotoxic responses are of types that could facilitate root infection.
In conclusion, I recommend accepting the manuscript for publication with minor revisions.
Other minor comments are reported here below.
L151: DS97-84-1 and Saline (Owen et al., 1994), susceptible to the disease
L170: 96 h ... hours must be abbreviated with 'h', but not 'hr', as reported in the International System of Units
L174: perhaps the term 'photobleaching' is not appropriate. You could consider changing into 'bleaching' or something else. Please check.
L149: report the experimental design
L175: how many replicates and plants were in the experiment? Only 3?
L186: report the number of plants, replicates, and the experimental design.
L187: which plant variety was used?
L202: report the number of plants, replicates, and the experimental design.
L187: which plant variety was used?
L218: which plant variety was used?  
L324: you could report such statistics in Fig4
L445: I would add a Conclusion section
Fig2: it is strange that 24h samples have SEM=0. It means that all samples had growth perfectly identical to the control, which is biologically impossible.
Fig4: report SEM and statistics
Fig6 should be placed prior to Fig5.
Fig7: control should be included in the picture
Fig8: make identical in size the left and right panels.

Author Response

Reviewer #3 felt that the part of the study measuring susceptibility to (±)-botryodiplodin of a charcoal rot resistant soybean, implied that botryodiplodin-related pathogenesis was related to the mechanism of resistance to charcoal rot, which it is not. Reviewer #3 recommended discussing it in the manuscript text, while avoiding emphasis on the fact that the phytotoxic responses are of types that could facilitate root infection.  This suggestion has been addressed in the revised manuscript by replacing the last phrase of the following sentence at lines 335 to 339: 

The observation that the three soybean genotypes examined in the study exhibited susceptibility to the phytotoxic effects of (±)-botryodiplodin in the order Saline > DS97-84-1 > DT97-4290 is consistent with the charcoal rot tolerance reported for genotype DT97-4290 (Paris et al., 2006) resulting from a change expressed in multiple tissues, including leaf tissue, rather than being a root-specific response associated with preventing initial infection.

with the following two sentences: 

The observation that the three soybean genotypes examined in the study exhibited susceptibility to the phytotoxic effects of (±)-botryodiplodin in the order Saline > DS97-84-1 > DT97-4290 is consistent with the charcoal rot tolerance reported for genotype DT97-4290 (Paris et al., 2006) resulting from a change expressed in multiple tissues, including leaf tissue. Given that the level of resistance expressed by genotype DT97-4290 is not sufficient to prevent charcoal rot disease and infection by M. phaseolina (Paris et al., 2006), subsequent studies focused on investigating root-specific responses believed to be associated with initial infection.

Reviewer #3 felt that the sentence at lines 150-151 could be improved by including that DS97-84-1 and Saline are susceptible to the disease. In the revised version this sentence now appears as follows:

Soybean genotypes DT97-4290 (moderately resistant to charcoal rot) (Paris et al., 2006), DS97-84-1 and Saline (Owen et al., 1994) (both of which are susceptible to charcoal rot) were grown in the greenhouse. 

Reviewer #3 pointed out that the appropriate abbreviation for hours is 'h', but not 'hr', as reported in the International System of Units. It has been changed throughout the manuscript.  Reviewer #3 felt the term 'photobleaching' is not appropriate, given that light effects were not studied. It has been changed to 'bleaching' throughout the revised manuscript. Reviewer #3 requested more detail on the experimental design for phytotoxicity assays. In response to this request we have added the following sentence at line 163, original manuscript:

Dose-response curves were obtained for phytotoxic responses to a range of (±)-botryodiplodin concentrations by triplicate cultures of three soybean leaf discs. 

And the following sentence at line 176, original manuscript:

Dose-response curves were obtained for phytotoxic responses to a range of (±)-botryodiplodin concentrations by triplicate cultures of three-frond duckweed plantlets. 

And the following sentence at line 203, original manuscript:

Dose-response curves were obtained for phytotoxic responses to a range of (±)-botryodiplodin concentrations by the roots of groups of three soybean seedlings cultured individually in hydroponic medium.  

Reviewer #3 felt the soybean variety should be mentioned on line 187 and 218. It is indicated as “commercial variety” on lines 188 and 218, original manuscript. Reviewer #3 suggested that statistics reported on lines 324-325 could be included in the legend to Fig 4. This has been done in the revised version by adding the sentence “Saline was significantly (P<0.05, multiple regression) more susceptible to the phytotoxic effects of (±)-botryodiplodin than DS97-84-1 and DT97-4290 at each time point.” at line 355, original version.  Reviewer #3 recommended adding a Conclusions section. The following section has been added at line 446, original manuscript:  Conclusions

The toxin, botryodiplodin, produced by M. phaseolina, the fungus that causes charcoal rot disease in many plant species, is phytotoxic in soybean leaf disc cultures and in actively growing Lemna pausicostata plantlet cultures.  Botryodiplodin exposed to undisturbed roots of soybean seedlings in hydroponic culture results in a root tip destruction response that would facilitate fungal infection of the root. 

Reviewer #3 asked about the lack of toxicity in 24 h samples in Fig2. In the revised manuscript a phrase has been added to line 301 in explanation as follows: 

No detectable toxicity was observed at 24 hours, because growth was measured as frond number and more time than that was needed for a plantlet to generate a new frond under conditions used. However, the full extent of toxicity was observed at 48 hours with IC50 = 0.22 μg/mL.

Fig4: The validity of doing statistics on percentage data was discussed above for Reviewer #2. Reviewer #3 felt Fig. 6 (the graphic results) should be placed prior to Fig. 5 (the picture of treated seedlings). This has been done in the revised version with appropriate changes in figure placement and text insertion of (Fig. 6).  Reviewer #3 felt Fig. 7 should include a control. In the experimental design, seedling roots treated with botryodiplodin in sand culture served as controls for unsupported seedling roots.  The legend to Fig. 7 in the revised manuscript has been revised to reflect this as follows: 

Figure 7.  Soybean seedling roots treated with a range of concentrations of (±)-botryodiplodin (10 to 300 µg/mL) in sand culture served as controls for unsupported soybean seedling roots in hydroponic culture.  A reduced number of lateral roots and discoloration occurs at higher concentrations of (±)-botryodiplodin.

In Fig. 8 at line 431, original manuscript, a figure has been inserted that is adjusted to make identical in size the left and right panels.

Reviewer 4 Report

In this manuscript, chemical synthesis of botryodiplodin was performed. A synthetic toxin has been used for antibacterial studies and plant tests (toxicity studies). The Reviewer considers that this work has certain importance in the field. However, in Reviewer opinion this work needs major improvements before considering its publishing in Toxins. What is more, some methodological and substantive issues need to be clarified and supplemented.

Some key problems that should be addressed by the Authors are discussed below:

Title: misleads readers and does not reflect analyses carried out by Authors in this work. In materials and methods section a description of botryodiplodin chemical synthesis was included. Whereas, the title informs that botryodiplodin was synthesized naturally by Macrophomina phaseolina and it is not true. Also, a suggestion that phytotoxic response is 'of type that could be used to facilitate root infection' is too far reaching. Mechanisms used by fungi to infect plant roots include not only secrection of toxins but also secretion of hydrolytic enzymes and mechanical damage caused by hyphae. What is more, the correct name of the fungus is Macrophomina phaseolina not Macrophomena phaseolina. Introduction Section: it is too long and chaotic and too many unnecessary information was provided, e.g. contamination of food caused by the toxin, development of soybean genotypes resistant to charcoal rot. Also, little information was provided about the latest studies (last 10 years) on botryodiplodin. In my opinion, the research goals (line 95-98) have been incorrectly formulated or not achieved. Authors claim that 'The objectives of the present study were to investigate the identity of the phytotoxin produced by M.  phaseolina isolates from Mississippi soybeans with charcoal rot disease as botryodiplodin'. I could not find the results (and methods) for this part of the research. Information provided in lines 125-130 is insufficient. lines 100-130, Preparation of botryodiplodin: there are no results confirming that the compound obtained is indeed a botryodiplodin. lines 248-254: Authors claim that purification of a natural botryodiplodin is impossible and those natural preparations contain highly toxic impurities. According to Authors application of a synthetic toxin is a better solution. However, purity of botryodiplodin used in this work is only 98% (line 124). How one can be sure that the remaining 2% is not toxic? There is no information on this. line 184-185, Fig. 2: What was 100% of inhibition? Please, explain. 8: What was the magnification for this micrograph? Please, provide the scale. Results and discussion section: In my opinion, discussion is a weak point of this work. Authors referred mainly to publication form 1960s-1980s and there is one extensive citiation of Ramezani et al., 2007. The results were not reliable and objectively discussed in the light of the latest research in the field. My objection is that based on several plant test (using synthetic toxin) Authors conclude about mechanism of root infection by M. phaseolina. Figures: According to instruction to Authors ' All Figures, Schemes and Tables should be inserted into the main text close to their first citation'. Fig 2, 3, 4, 5, and 7 do not meet this guidelines. There are numerous editorial errors: font size e.g.: line 68, 118, 255.

Therefore, the Reviewer suggests MAJOR manuscript correction.

Author Response

Reviewer #4 felt that the title was not reflective of the analyses carried out by Authors in this work and that a suggestion in the title that phytotoxic response is 'of type that could be used to facilitate root infection' is too far reaching, specifically that it excluded the possibility that hydrolytic enzymes and mechanical damage caused by hyphae could also be involved in the infection mechanism. Reviewer #4 also pointed out a typographical error in the name of the fungus.

In the revised manuscript the title “Botryodiplodin, a toxin produced by the charcoal rot disease fungus, Macrophomena phaseolina, induces phytotoxic responses in soybean (Glycine max L.) that are of types that could be used to facilitate root infection” has been replaced by “Phytotoxic responses of soybean (Glycine max L.) to botryodiplodin, a toxin produced by the charcoal rot disease fungus, Macrophomina phaseolina”.  We feel that this revised title will not be interpreted as excluding the possibility that hydrolytic enzymes and mechanical damage caused by hyphae could also be involved in the infection mechanism. 

Reviewer #4 felt that the Introduction Section is too long and unfocussed, specifically requesting that material on contamination of food caused by the toxin, and on development of soybean genotypes resistant to charcoal rot be eliminated. In the revised manuscript these subjects have been deleted by removing the following text:  45-48, original manuscript:

Because M. phaseolina is known to produce (-)-botryodiplodin and to be present in seeds as an endophyte, contamination of food items such as tofu and vegetable oil by (-)-botryodiplodin is a concern (Shier et al., 2007). However, studies on foods and feeds impacted by charcoal rot disease have not been reported.

55-59, original manuscript:

Extensive studies have been carried out attempting to use selective breeding to develop soybean genotypes that are resistant to charcoal rot, but this approach has yielded only tolerant or moderately resistant genotypes (Mengistu et al., 2009; Paris et al., 2006). Attempt to use various agronomic techniques to prevent the disease have also failed, so research on charcoal rot continues (Bellaloui et al., 2012; Mengistu et al., 2009).

As a result the reference Mengistu et al., 2009 has been removed.

Reviewer #4 requested additional detail on the characterization of the (±)-botryodiplodin produced beyond the information on NMR spectral characterization and chromatographic characteristics in ll. 125-130 as follows:

The (±)-botryodiplodin (6) exhibited 1H nuclear magnetic resonance spectroscopy values and thin layer chromatographic Rf values identical to those reported in the literature (Nouguier et al., 2003; Rehnberg & Magnusson, ‎1990) and those obtained in this laboratory with (-)-botryodiplodin purified from cultures of M. phaseolina (Ramezani et al., 2007), except that (±)-botryodiplodin was not optically active.

While NMR spectral characterization is rightly or wrongly the standard in the field, we are happy to include the following additional material after l. 130, original manuscript.  It was not included in the original manuscript to save space, given that it is all conventional methodology in the field. 

The synthetic (±)-botryodiplodin was also shown to exhibit the following properties indistinguishable from (-)-botryodiplodin of natural origin: liquid chromatography/mass spectral characteristics, antibacterial activity (Sen Gupta et al., 1966) and delayed formation of a red pigment on inadvertent contact with skin (Sen Gupta et al., 1966; McCurry & Abe, 1973). 

Reviewer #4 apparently felt the rationale given in lines 248-254 for using chemical synthesis to provide evidence that toxicity in a preparation purified from nature does not reside in an impurity was over-stated. Specifically, how can one be sure that a 2% impurity in the synthetic toxin is not toxic? In the revised manuscript the passage has modified as follows: 

Chemical synthesis of a toxin is unlikely to produce the same impurities as found in material purified from nature.  Even if the impurities in the two types of preparations are both toxic, they are unlikely to induce identical pathology in all toxicity tests. Therefore, identical phytotoxic properties are unlikely to be observed in synthetic and natural preparations of a toxin, if the activities of either are due to a highly active impurity.

Reviewer #4 requested clarification of how 100% inhibition could occur with duckweed plantlet growth on line 184-185. The description given is indeed unclear and it has been replaced with the following line in the revised manuscript: 

Growth was measured as addition of fronds in treated cultures relative to control cultures not treated with (±)-botryodiplodin.  No additional fronds in a treated culture was scored as 100% inhibition of growth.   

The magnification for the micrographs in Figure 8 have been provided in the revised manuscript at lines 433 and 434, original manuscript as follows:

“. . . (left panel, 400X) or with (±)-botryodiplodin (15 µg/mL) (right panel, 200X).”

Reviewer #4 raised concerns nearly the same as by Reviewer #2, item 2, above, namely “Authors referred mainly to publication form 1960s-1980s and there is one extensive citation of Ramezani et al., 2007. The results were not reliable and objectively discussed in the light of the latest research in the field.” As discussed above, we are pleased to add the citation of the accompanying manuscript, Abbas et al. (2019) to the revised version of this manuscript. As indicated above, botryodiplodin is in reality a neglected toxin, with the result there is a lack of other relevant references appearing in the last 5 years. We hope our discoveries on the roles of botryodiplodin in root infection will change that.  Reviewer #4 had an objection about conclusions drawn. It is expected that the 4. Conclusions section added following line 446, original manuscript, in response to a suggestion by Reviewer #3, item 8, above will address this concern. Reviewer #4 expressed concern that according to instruction to Authors, “All Figures, Schemes and Tables should be inserted into the main text close to their first citation”. Reviewer #4 felt that Figs. 2, 3, 4, 5, and 7 do not meet this guidelines. Given that these were placed by the editorial staff, we are willing to defer to them about Fig. placement. Reviewer #4 also indicated there are numerous editorial errors, notably font size e.g.: line 68, 118, 255.  These and other font issues have been corrected in the revised manuscript.

The following statement has been added for the Author Contributions section on line 446, original manuscript: 

Conceptualization, Hamed Abbas and W. Thomas Shier; Formal analysis, Alemah Butler and W. Thomas Shier; Funding acquisition, Hamed Abbas; Investigation, Hamed Abbas, Nacer Bellaloui, Alemah Butler, Justin Nelson, Mohamed Abou-Karam and W. Thomas Shier; Project administration, Hamed Abbas; Writing – original draft, Hamed Abbas; Writing – review & editing, Nacer Bellaloui, Alemah Butler, Justin Nelson, Mohamed Abou-Karam and W. Thomas Shier.

Round 2

Reviewer 1 Report

The manuscript was improved after the revision process.

In my opinion, the sentence reported by the authors in the revised version (Lines 283-286) is inexact.

“More extensive structural alterations of botryodiplodin in the form of epimers have been reported to be inactive in the case of 4-epi-botryodiplodin (Fujimoto et al., 1980) and almost inactive in the case of 3-epi-botryodiplodin  (Félix et al., 2019)”

In particular, about 3-epi-botryodiplodin, authors report that this metabolite was almost inactive, whereas in Felix et al 3-epi is more phytotoxicity on tomato leaves then botryodiplodin. Please review this concept.

Furthermore, Fujimoto et al., 1980 and Felix et al 2019 are not included in the reference list

Author Response

Reviewer #1 pointed out that the discussion in the revised manuscript on toxicity of 3-epibotryodiplodin addressed only mammalian toxicity reported in Félix et al. (2019). The following expanded discussion of 3-epi-botryodiplodin has been introduced into the current version of the manuscript at lines 289-293:

More extensive structural alterations of botryodiplodin in the form of epimers have been reported to be inactive in the case of 4-epi-botryodiplodin (Fujimoto et al., 1980). Félix et al. (2019) observed that cytotoxicity of 3-epi-botryodiplodin measured in Vero monkey kidney cells and 3T3 mouse fibroblast cultures was 0-5% of the cytotoxicity of botryodiplodin. However, in a leaf puncture assay in young tomato plant leaves, 3-epi-botryodiplodin produced a much larger lesion with different morphology than botryodiplodin, but similar to the lesion produced by botryodiplodin acetate.

2. Two types of reference problems have been corrected in this version of the manuscript. First, the following two references were given in the cover letter and stated as added to the manuscript in response to suggestions from Reviewer #1, but in fact through an oversight did not get added. They have been added to the current manuscript version.

12. Félix C, Salvatore MM, DellaGreca M, Ferreira V, Duarte AS, Salvatore F, Naviglio D, Gallo M, Alves A, Esteves AC, Andolfi A. 2019. Secondary metabolites produced by grapevine strains of Lasiodiplodia theobromae grown at two different temperatures. Mycologia, 111:466-476.

13. Fujimoto Y, Kamiya M, Tsunoda H, Ohtsubo K, Tatsuno T. 1980. Recherche toxicologique des substances métaboliques de Penicillium carneo-lutescens. Chemical and Pharmaceutical Bulletin, 28:1062-1066.

Secondly, the reference to the accompanying paper, Abbas et al., 2019, was incorrectly placed in presentation order rather than alphabetically. In the present version of the manuscript, it has been moved to alphabetical order as follows:

1. Abbas HK, Bellaloui N, Accinelli C, Smith JR, Shier WT. 2019. Toxin production in soybean (Glycine max L.) plants with charcoal rot disease and by Macrophomina phaseolina, the fungus that causes the disease. Toxins 11:645; doi:10.3390/toxins11110645.

Reviewer 4 Report

The manuscript 'Phytotoxic responses of soybean (Glycine max L.) to botryodiplodin, a toxin produced by the charcoal rot disease fungus, Macrophomina phaseolina' has been significantly improved and now is suitable for publication.

Author Response

Reviewer $; no additional revisions requested